# Thermal Treatment Impact on the Mechanical Properties of Mg_3_Si_2_O_5_(OH)_4_ Nanoscrolls

**DOI:** 10.3390/ma15249023

**Published:** 2022-12-16

**Authors:** Andrei Krasilin, Maksim Khalisov, Ekaterina Khrapova, Valery Ugolkov, Andrey Enyashin, Alexander Ankudinov

**Affiliations:** 1Ioffe Institute, 194021 St. Petersburg, Russia; 2Pavlov Institute of Physiology of the RAS, 199034 St. Petersburg, Russia; 3Grebenshchikov Institute of Silicate Chemistry of the RAS, 199034 St. Petersburg, Russia; 4Institute of Solid State Chemistry UB RAS, 620108 Ekaterinburg, Russia

**Keywords:** phyllosilicate, nanotube, hydrothermal synthesis, atomic force microscopy, thermal analysis, density functional theory, Young’s modulus, shear modulus

## Abstract

A group of phyllosilicate nanoscrolls conjoins several hydrosilicate layered compounds with a size mismatch between octahedral and tetrahedral sheets. Among them, synthetic Mg_3_Si_2_O_5_(OH)_4_ chrysotile nanoscrolls (obtained via the hydrothermal method) possess high thermal stability and mechanical properties, making them prospective composite materials fillers. However, accurate determination of these nano-objects with Young’s modulus remains challenging. Here, we report on a study of the mechanical properties evolution of individual synthetic phyllosilicate nanoscrolls after a series of heat treatments, observed with an atomic force microscopy and calculated using the density functional theory. It appears that the Young’s modulus, as well as shear deformation’s contribution to the nanoscrolls mechanical behavior, can be controlled by heat treatment. The main reason for this is the heat-induced formation of covalent bonding between the adjacent layers, which complicate the shear deformation.

## 1. Introduction

Determining the mechanical characteristics of micro- and nano-objects is a challenging but practically important task. The results of individual particles studies are in demand both for predicting the mechanical properties of composite materials based on them [1,2,3], and in various local diagnostics methods [4], including those applied for living systems [5,6,7]. This stimulates the development of new and existing measurement methods, such as atomic force microscopy (AFM) [8] and nanoindentation [9,10].

To date, there are several morphological types of nanoparticles with specific mechanical properties, including rods [4,11,12,13], tubes and scrolls [14,15,16], as well as onion-based compounds with a layered structure [17,18,19]. The amount of attention paid to this group has dramatically increased since evidence of an unusually high value of Young’s modulus—about 1 TPa—of carbon nanotubes was obtained [20]. Within this context, phyllosilicate nanoscrolls with chrysotile [21,22,23] and halloysite structures [24,25] are as interesting as their carbon analogues. The nanotubular morphology of these hydrosilicates originates from the size mismatch between the metal–oxygen and silicon–oxygen sheets [14]. The average values of Young’s modulus of phyllosilicate nanoscrolls obtained by performing AFM [21,22] and quantum chemical calculations [23,24] are in the range of 150–300 GPa. High mechanical characteristics, heat resistance, the presence of an inner channel (unlike most other silicates), and an abundance of OH groups on the surface ensure the widespread application of phyllosilicates as fillers of functional composite materials [14,26,27,28,29] as well as catalyst supports [30,31,32] and nanocontainers [33,34,35]. In comparison with halloysite, chrysotile nanoscrolls of various compositions can be easily synthesized using a hydrothermal process [14].

Experimental studies of the mechanical behavior of nano-objects by AFM bending tests still proceed with difficulties of a methodological and physical nature. The first group is associated with the uncertainty in AFM cantilever behavior, the conditions of probe-sample contact, and nano-object boundary conditions, which are unknown a priori. The registration of the bending angle and the torsion angle of the cantilever using the traditional AFM scheme based on the optical lever does not provide control over all three spatial components of the probe–sample interaction force. As a result, there is no information as to whether the probe is clamped on the surface, or if it slides along the surface of the nano-object. This leads to a discrepancy between the measured stiffness signal and the real contact stiffness [36]. Moreover, even with the correct determination of the contact stiffness, the unknown boundary conditions of the nano-object may be a reason for, theoretically, an unlimited underestimation of the Young’s modulus during the AFM bending test.

Attempts to account for these factors actualized the issues of the applicability of the Bernoulli–Euler theory for the case of the bending behavior of nanorods and nanotubes. Within the framework of this theory, it was impossible to describe the growth of the Young’s modulus with a decrease in the outer diameter, which was observed for nanorods, nanotubes, and various phyllosilicate nanoscrolls [11,12,13,25]. Previous studies considered the possible contribution of surface tension [13] or shear deformations [25] to the deflection of a suspended nano-object. The contribution of surface tension, which increased with a decrease in the length-to-diameter ratio (aspect ratio), led to an overestimation of the measured Young’s modulus. On the contrary, the contribution of shear strain, which increased with a decrease in the aspect ratio, yielded an underestimation of the measured value. It is worth adding, however, that a recent study [21] indicated an increase in the scatter, rather than in the value of the Young’s modulus itself, with a decrease in the nano-object diameter.

According to thermal analysis [37,38,39,40], phyllosilicate nanoscrolls retained their nanotubular morphology at temperatures of up to at least 600 °C, despite the occurrence of structural transformations associated with the surface and volume dehydroxylation of particles. Here, we attempted to use this feature to initiate covariation in the mechanical properties of individual Mg_3_Si_2_O_5_(OH)_4_ phyllosilicate nanoscrolls to further clarify the size dependence of the Young’s modulus on the outer diameter. The temperature evolution of the mechanical behavior of each particle was traced using the AFM method, and the experiment was compared with the simulation of mechanical properties within the framework of the density functional theory (DFT).

## 2. Materials and Methods

### 2.1. Nanoscrolls Synthesis and Analysis

Mg_3_Si_2_O_5_(OH)_4_ phyllosilicate nanoscrolls were obtained by performing hydrothermal treatment of the initial composition in line with the paper [21]. To prepare the initial composition, a sample of amorphous SiO_2_ (Aerosil A-300) was dissolved in a 2M NaOH aqueous solution at constant stirring for 24 h. Then, a 1M MgCl_2_ aqueous solution was added dropwise to the mixture. The amounts of MgCl_2_, SiO_2_, and NaOH were calculated in such a way as to ensure (a) compliance with Mg_3_Si_2_O_5_(OH)_4_ stoichiometry, and (b) complete precipitation of MgCl_2_. The resulting precipitate was washed from the side products of the precipitation reaction with distilled water by performing repeated centrifugation, dried in air at 90 °C, and ground in an agate mortar.

The hydrothermal treatment was carried out in a 400 mL stainless steel autoclave (REXO Engineering, Seoul, South Korea). After adding 1 g of the initial composition, as well as 200 mL of a 0.1M NaOH aqueous solution, the autoclave was sealed and held at a temperature of 353 °C and a pressure of around 22 MPa for 10 h. After cooling, the hydrothermal treatment product was washed with distilled water until it reached a neutral pH level, and dried in air at 90 °C. To determine the change in the phase composition, two parts of the hydrothermal treatment product were subjected to heat treatment in air for 8 h at temperatures of 400 and 600 °C.

The phase composition of the hydrothermal and heat treatment products was studied by performing powder X-ray diffractometry (PXRD, Rigaku SmartLab 3 with copper anode and $K_{\beta }$ filter, Rigaku Corporation, Tokyo, Japan). The phases were identified using the ICDD PDF-2 database. The element composition was studied using energy-dispersive X-ray spectroscopy (EDS) as part of the scanning electron microscopy method (SEM, FEI Quanta 200, FEI Company, Hillsboro, OR, USA). A synchronous thermal analysis of the product of hydrothermal treatment was carried out according to a method presented in [37]. The Netzsch STA 429 CD analyzer was used in differential scanning calorimetry and the thermogravimetry (DSC-TG) mode. During the analysis, a sample in the form of a 20 mg tablet in a corundum crucible was heated from 40 to 1100 °C at a rate of 20°/min in the air flow.

### 2.2. Sample Preparation for the AFM Experiment

To study the mechanical behavior of individual nanoparticles, a drop of phyllosilicate nanoscrolls suspension in isopropanol after preliminary ultrasonic dispersion (about 20 min) was applied to the preheated (60 °C) substrate to speed up the drying process and prevent aggregation. The substrate was a TGZ2 (NT-MDT SI, Zelenograd, Moscow, Russia) calibration grid of Si/SiO_2_ with regular grooves of about 110 nm deep and 1500 nm wide. The SEM method was used to localize about 100 phyllosilicate nanoscrolls that overlap the grooves on the substrate (Figure 1)—the so-called nanobridges [21]—some of which were used in further experiments. After the first series of AFM experiments, the substrate with nanobridges was kept in an argon flow (to prevent silicon oxidation) for 8 h at temperatures of 400 °C; after the second series of AFM experiments, the substrate was kept at 600 °C. Thus, changes in the mechanical behavior of the same individual nanobridges were traced with an increase in the temperature of the heat treatment.

### 2.3. AFM Experiment and Its Treatment

AFM experiments were carried out according to the method described in detail elsewhere [21,36,41]. The BioScope Catalyst (Bruker, Santa Barbara, CA, USA) atomic force microscope was used to test the mechanical properties of the nanoscrolls. The Z16 APO (Leica, Wetzlar, Germany) optical microscope integrated with the atomic force microscope helped to identify nanobridges previously localized by SEM. Bending tests were performed using FMG01 cantilevers (NT-MDT SI, Zelenograd, Moscow, Russia) with an average spring constant of 3 N/m according to the manufacturer’s specification. The spring constant of each cantilever was characterized by the thermal tune method before use [42]. Bending tests of nanobridges were carried out in the PeakForce QNM AFM operation mode, in which the relief of the sample surface is visualized and the AFM force curves are analyzed simultaneously. The peak force setpoint value was chosen individually for each nanoscroll to obtain the maximum nanobridge deflection of about 10 nm.

AFM data were processed using the Gwyddion 2.58 program (Czech Metrology Institute, Brno, Czech Republic) [43]. To consider methodological factors, the deformation maps of nanobridges obtained as a result of AFM imaging, δ, were corrected [21,36]. In addition to the deformation map, the AFM peak force error $F_{E}$ and height images were also used for correction. The corrected deformation $\delta_{C}$ was calculated according to the following expression:(1)$$\delta_{C}=\frac{\delta }{0.85}\frac{F_{\mathrm{SP}}}{F_{\mathrm{SP}}+F_{E}}\cos^{2}\theta$$
where $F_{\mathrm{SP}}$ is the peak force setpoint and θ is the angle between the normal to the sample surface and the vertical.

The minimum bending stiffness of the nanobridge was determined using the following Equation:(2)$$k_{\exp}=\frac{F_{\mathrm{SP}}}{\delta_{C}}$$

Two deflection profiles of different lengths were extracted from the AFM map of corrected deformation (Figure 1). The first one was the length of the suspended part of the nanoscroll according to the AFM height image. The second one was the length of the curve fragment, where the deformation signal was different from zero (i.e., the length was determined based on the AFM deformation map). Further, both profiles were normalized along two axes and approximated according to the dependence ζ(χ); the mechanical behavior of the nanobridge was theoretically predicted as follows:(3)$$\zeta \left(\chi \right)=4^{n}{\left(\chi -\chi^{2}\right)}^{n}$$
where the parameter n reflects the nanobridge boundary conditions (n = 2 for a supported beam, n = 3 for a clamped beam).

A deformation profile with a more accurate agreement of the theoretical and experimental curves was selected from two profiles. The selected profile was approximated by one of the following expressions, depending on the assumed properties of the foundation. For a rigid foundation, the following expression was used:(4)$$\zeta_{\lambda }\left(\chi \right)=32\frac{3\lambda \left(\lambda +1\right){\left(\chi -\chi^{2}\right)}^{2}+2\left(\lambda +2\right){\left(\chi -\chi^{2}\right)}^{3}}{\left(2\lambda +1\right)\left(3\lambda +2\right)}$$
where λ is the fitting parameter.

In the case of an elastic foundation [44], we used the following expression:(5)$$\begin{array}{c} \zeta_{\beta }\left(\chi \right)=32\frac{3\left(6+12\beta +12\beta^{2}+6\beta^{3}+\beta^{4}\right)+6\left(3+4\beta +\beta^{2}\right)\beta^{3}\left(\chi -\chi^{2}\right)}{\left(2+\beta \right)\left(12+12\beta +6\beta^{2}+\beta^{3}\right)\left(24+12\beta +6\beta^{2}+\beta^{3}\right)}\\ +32\frac{6\left(1+3\beta +\beta^{2}\right)\beta^{4}{\left(\chi -\chi^{2}\right)}^{2}+2\left(2+\beta \right)\beta^{6}{\left(\chi -\chi^{2}\right)}^{3}}{\left(2+\beta \right)\left(12+12\beta +6\beta^{2}+\beta^{3}\right)\left(24+12\beta +6\beta^{2}+\beta^{3}\right)} \end{array}$$
where β is the fitting parameter.

The correction parameters of the fixing conditions for the two types of foundations were determined, respectively, with the following Equations:(6)$$\Phi \left(\lambda \right)=1+\frac{3\lambda }{\lambda +2}$$
and
(7)$$\Phi \left(\beta \right)=1+\frac{24+12\beta +6\beta^{2}}{\beta^{3}}$$

The Young’s modulus $E_{\mathrm{cor}}$, corrected for fixing conditions, was determined according to the case of bending of a clamped cylindrical beam with modulus $E_{\mathrm{CB}}$:(8)$$E_{\mathrm{cor}}=[\begin{array}{c} \Phi \left(\lambda \right)E_{\mathrm{CB}}\\ \Phi \left(\beta \right)E_{\mathrm{CB}} \end{array};\;E_{\mathrm{CB}}=\frac{k_{\exp}l^{3}}{3\pi D^{4}}$$
where l is the length chosen based on Equation (3) and *D* is the outer diameter of the nanoscroll.

It was shown in [21] that one of the most likely reasons for the spread in Young’s modulus values was the contribution of shear deformation to the complex mechanical behavior of the nanoscroll, which lowered the observed Young’s modulus compared to its true value.

According to the theory of elasticity [15,25,45], the shear modulus value can be determined as a result of the linear approximation of experimental data with the following Equation:(9)$$\frac{1}{E_{\mathrm{cor}}}=\frac{10D^{2}}{3\Phi \left(\beta \right)l^{2}}\frac{1}{G}+\frac{1}{E_{G}}$$
where G is the shear modulus and $E_{G}$ is the shear-corrected Young’s modulus.

The value of the parameter β was also used [44] to estimate the stiffness of the elastic foundation:(10)$$k_{W}=\frac{\pi \beta^{4}D^{4}E_{\mathrm{cor}}}{16l^{4}}$$

### 2.4. DFT Modeling of Mechanical Properties

The DFT calculations within periodic boundary conditions were carried out using the SIESTA 4.0 package (MaX Center of Excellence, Modena, Italy) [46,47]. Generalized Gradient Approximation (GGA) with Perdew–Burke–Ernzerhof (PBE) parametrization was employed to determine the exchange–correlation potential. The core electrons were treated within the frozen core approximation, applying norm-conserving Troullier–Martins pseudopotentials. The valence orbitals were described using the single-ζ basis set for Mg, Si, and H elements and the double-ζ basis set for O. The chosen basis set is not inferior to the global double-ζ basis in the description of crystallographic and mechanical properties of chrysotile with a visible benefit in computational time. The k-point mesh was generated using the method of Monkhorst and Pack with a cutoff of 15 Å used for k-point sampling [48]. Effectively, it corresponded to the k-point sampling grids 6 × 4 × 3 for the chrysotile lattice and 3 × 2 × 6 for the sepiolite lattice. The real-space grid used for the numerical integrations was set to correspond to the energy cutoff of 300 Ry. The calculations were performed using atomic position relaxations and cell optimization, when required, with convergence criteria corresponding to the maximum residual stress of 0.1 GPa for each component of the stress tensor and the maximum residual force component of 0.05 eV/Å.

As an initial setup of the unit cells of chrysotile and sepiolite, the lattice parameters and the atomic positions were adopted from the X-ray powder diffraction data for tubular-shaped dry chrysotile [49] and for hydrated sepiolite [50]. The unit cell of chrysotile with a stoichiometry of Mg_3_Si_2_O_5_(OH)_4_ contained 72 atoms. The unit cell of sepiolite contained 152 atoms and had a nominal composition of Mg_4_Si_6_O_15_(OH)_2_·3H_2_O; the water molecules with the experimentally poorly defined positions and with occupancy numbers of O atoms below 0.5 were not accounted for. The equilibrium lattice parameters of both chrysotile and sepiolite demonstrate a fair reliability of the chosen DFT scheme for reproduction of the experimental lattice parameters within ±2% error. For chrysotile, theoretical data a = 5.33 Å, b = 9.23 Å, c = 14.47 Å, β = 91.1° were fitted to experimental data a = 5.340 Å, b = 9.241 Å, c = 14.689 Å, β = 93.66° [49]. For sepiolite, theoretical data a = 12.75 Å, b = 27.43 Å, c = 5.32 Å were comparable to experimental data a = 13.405 Å, b = 27.016 Å, c = 5.275 Å [50], but the a-parameter was underestimated. Seemingly, this underestimation unveiled a swelling of real sepiolite by loose water molecules in channels of the lattice, which were overlooked in our theoretical model.

Young’s modulus $E_{\mathrm{DFT}}$ was calculated using the following Equation:(11)$$E_{\mathrm{DFT}}=\frac{1}{V}{\left(\frac{\partial^{2}U}{\partial \varepsilon^{2}}\right)}_{\varepsilon =0}$$
where V is the volume of the init cell at equilibrium, ε is the strain along chosen lattice vector, and U is the calculated total energy of the stressed unit cell.

The shear modulus $G_{\mathrm{DFT}}$ was calculated using the following Equation:(12)$$G_{\mathrm{DFT}}=\left[\frac{1}{\Delta x}\left(\frac{\partial U}{\partial \Delta x}\right)\right]\frac{l_{c}}{A}$$
where Δx is the shear displacement, A is the area of unit cell experiencing the shear force action, and $l_{c}$ is the initial length of unit cell, which is orthogonal to displacement. The values of A, $l_{c}$, and V were kept constant at different displacements and corresponded to those at equilibrium.

## 3. Results and Discussion

### 3.1. Phase Transitions during the Heat Treatment

The heat treatment of Mg_3_Si_2_O_5_(OH)_4_ in the 40–1100 °C temperature range initiated several processes accompanied by thermal effects and mass losses (Figure 2). The endothermic effect at around 100 °C led to a weight loss of about 4%, which was due to the removal of adsorbed water. The magnitude of the effect indirectly confirmed the large specific surface area of the hydrothermal treatment products due to their nanotubular nature. A further increase in temperature initiated dehydroxylation of the phyllosilicate outer surface, gradually affecting layers deeply in the nanoscroll wall. Due to this, the endothermic effect of this process was relatively wide.

Since the hydroxyl groups ensured that the distance between adjacent layers of the chrysotile structure (Figure 3) was maintained at the length of hydrogen bonds, their removal led to the partial attachment of the adjacent layers to each other with the formation of sepiolite-like phase [31,51] and amorphous phase, according to the following reaction:(13)$$3{\mathrm{Mg}}_{3}{\mathrm{Si}}_{2}O_{5}{\left(\mathrm{OH}\right)}_{4}\to {\mathrm{Mg}}_{4}{\mathrm{Si}}_{6}O_{15}{\left(\mathrm{OH}\right)}_{2}+5{\mathrm{MgO}}_{\left(\mathrm{am}.\right)}+5H_{2}O$$

The Mg:Si ratio in sepiolite layers (2:3) is the opposite to that found in the chrysotile layers (3:2); therefore, part of the layer material was amorphized, leading to an additional contribution to the endothermic effect and disturbing the crystal structure (Figure 4) of the sample. Dehydroxylation occurred partially because it was necessary to ensure the structural stability of the sepiolite-like phase. Further weight loss in the 650 to 800 °C temperature range (Figure 2) was caused again by dehydroxylation, but of the sepiolite-like phase. A sharp exothermic effect at temperatures of above 800 °C corresponded to the crystallization of magnesium silicates, mainly in the form of Mg_2_SiO_4_ forsterite [37].

### 3.2. Alteration of Individual Nanobridge Parameters

Figure 1 shows an example of AFM data obtained from a nanoscroll exposed to heat treatment. It can be seen that, because of thermal treatment, the fitting parameter β decreased, while the corrected value of Young’s modulus, on the contrary, increased. The heat treatment of the substrate with each sample led to a change in both the size parameters and mechanical characteristics of the nanoscrolls. Figure 5 shows the tracks of the main characteristics of nanobridges. The bending stiffness of nanobridges increased relatively weakly with increasing treatment temperature, while the outer diameter (included in Equation (8) as $D^{-4}$) decreased due to the dehydroxylation of the outer surface and nanoscroll walls.

A change in the boundary conditions contributed significantly to the final $E_{\mathrm{cor}}$ value—most of the nanobridges demonstrated mechanical behavior closer to the supported beam type with an increase in the heat treatment temperature. The transition to such a state could be associated with a significant weakening of the adhesion to the substrate after dehydroxylation of the nanobridge surface. It should be noted that it was necessary to use the Φ(β) parameter, which takes into account the boundary conditions of nanobridge situated on an elastic foundation, instead of Φ(λ) (the latter was obtained for the case of a foundation with infinite stiffness). A foundation with finite stiffness $k_{W}$ (see inset in Figure 6), might be a consequence of the partial amorphization of either the substrate or the nanoscrolls themselves (Figure 4).

### 3.3. Changes in Mechanical Behavior

The heat treatment of phyllosilicate nanoscrolls yielded an almost two times increase in the mean Young’s modulus (inset in Figure 6) due to the synergistic effect of the increase in bending stiffness, in the value of Φ(β) parameter and the decrease in the outer diameter. The main increase occurred after treatment at 600 °C, which highlighted the key contribution of the phase transition to the sepiolite-like phase to the mechanical behavior of nanobridges. The dependence of the Young’s modulus on the outer diameter was observed in a series of papers [11,12,13,21,25] devoted to the study of the mechanical characteristics of individual anisotropic nanoparticles (rods, tubes, scrolls). Despite the size dependence that was also observed in this study, the heat treatment almost vanished it. Thus, the Young’s modulus of the nanobridges treated at 600 °C could be successfully fitted with a horizontal straight line within the measurement error in the coordinates of Figure 6. The inset in Figure 6 also shows the results of the $k_{W}$ value estimation. The decrease in the $k_{W}$ value with an increase in heat treatment temperature by more than an order of magnitude was probably due to partial amorphization of the crystal structure (Figure 4).

The role of heat treatment may be most clearly demonstrated by linearizing the experimental data according to Equation (9). Figure 7a shows that the degree of data linearity increased significantly with an increase in the heat treatment temperature. All points could not be approximated by a straight line with a single pair of $E_{G}$ and G values until treatment at 600 °C. A probable reason for this issue is the variation in the direction of scrolling found for phyllosilicate nanoscrolls using computational [53,54] and experimental methods [55]. This variation provided the basis for the formation of the shear modulus distribution due to different conditions of hydrogen bonding of the adjacent phyllosilicate layers. According to [21], the average shear modulus of the Mg_3_Si_2_O_5_(OH)_4_ nanoscrolls was 1.5 GPa, which was used to approximate the data in the region before the transition to the sepiolite-like phase (including 400 °C, see Figure 7). Although this made it possible to describe some of the points, it was necessary to use different values of DFT which were calculated with shear and Young’s moduli (Table 1). It should be noted that the difference between the Young’s and the shear moduli obtained with the DFT method along different directions may be considered as the difference in the mechanical characteristics of the layers scrolled along the [100] or [010] directions. Thus, before the phase transition to the sepiolite-like phase, the mechanical characteristics of individual nanobridges can be described using Equation (9), the results of DFT modeling, and the previously estimated [21] average shear modulus.

The phase transition during heat treatment at 600 °C led to an increase in the average $E_{\mathrm{cor}}$ value (Figure 6). In the linear coordinates of Equation (9), the experimental points lined up along one line with a relatively small slope (Figure 7), which indicated a significant increase in the shear modulus of individual nanobridges. The DFT modeling results showed (Table 1) that sepiolite possessed a larger average shear modulus than chrysotile. We believe that the increase in the shear modulus was caused by a change in the nature of the chemical bonding between adjacent layers, whereby hydrogen bonding in the chrysotile structure was replaced by Si-O-Si oxygen bridges in the sepiolite structure (Figure 2). The Si-O covalent bond significantly exceeds the hydrogen bond in terms of energy, so it is much more difficult to deform it. Thus, new chemical bonds arising during thermal treatment prevented shear deformations in the nanobridges, yielding an increase in Young’s modulus $E_{\mathrm{cor}}$ and the degree of experimental data linearization.

An attempt to approximate the points in Figure 7d with the $E_{\mathrm{DFT}}$ and $G_{\mathrm{DFT}}$ values obtained from Table 1, however, was not successful due to the lower Young’s modulus calculated for sepiolite compared to the observed $E_{\mathrm{cor}}$ values. The main reason for the discrepancy between the DFT calculated and experimental values may be the use of a sepiolite cell in its native form, while the real structure of the nanoscroll is complicated by the effects of thermal decomposition—dehydroxylation and the formation of amorphous substance in the wall according to the reaction (13). Filling the cavities of the sepiolite structure with an amorphous substance similar to MgO (Figure 3) could significantly affect the calculated Young’s modulus of such composite.

## 4. Conclusions

A unique series of AFM experiments was carried out to observe the heat treatment impact on the mechanical behavior of individual nanobridges based on Mg_3_Si_2_O_5_(OH)_4_ phyllosilicate nanoscrolls with a chrysotile structure. To conclude, heat treatment initiated simultaneous increases in the Young’s modulus and the shear modulus of the nanoscrolls due to phase transition to the sepiolite-like phase. Prior to the transition, the observed values of the Young’s modulus were affected by a significant contribution of shear strain due to hydrogen bonding between adjacent layers and, hence, a relatively low shear modulus value. Shear strain was also responsible for the variations in the measured Young’s modulus value. The phase transition joined the adjacent phyllosilicate layers using Si-O-Si bonds (instead of hydrogen bonds), preventing shear strain. The proposed mechanism of mechanical properties alteration was consistent with the results of the DFT modeling of chrysotile and sepiolite mechanical properties. Thus, the mechanical behavior of phyllosilicate nanoscrolls can be controlled by performing heat treatment.

## Figures and Tables

**Figure 1 materials-15-09023-f001:**
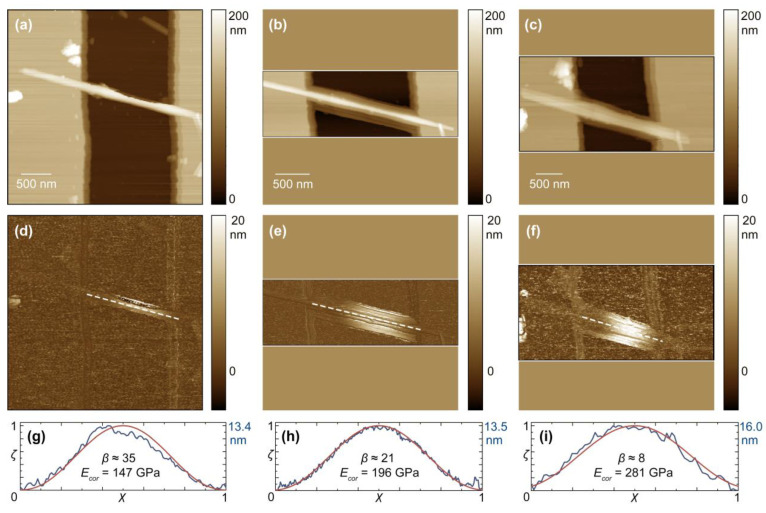
AFM (**a**–**c**) height and (**d**–**f**) corrected deformation images of the Mg_3_Si_2_O_5_(OH)_4_ nanoscroll on a TGZ2 silicon grating before treatment, after heat treatment at 400 °C;, and after heat treatment at 600 °C, respectively; (**g**–**i**) normalized profiles of corrected deformation of the nanoscroll (blue) and theoretical profiles according to the expression (5) (red), the extraction sites of the profiles are shown by dashed lines on (**d**–**f**), respectively. AFM imaging was performed with Peak Force Amplitude and Frequency—150 nm and 1 kHz, respectively. The other scanning parameters were as follows: (before heat treatment) Peak Force Setpoint—20 nN, Scan Rate—0.3 Hz, Samples/Line—256, Lines—256; (400 °C) 25 nN, 0.3 Hz, 512, 171; (600 °C) 40 nN, 0.2 Hz, 256, 124.

**Figure 2 materials-15-09023-f002:**
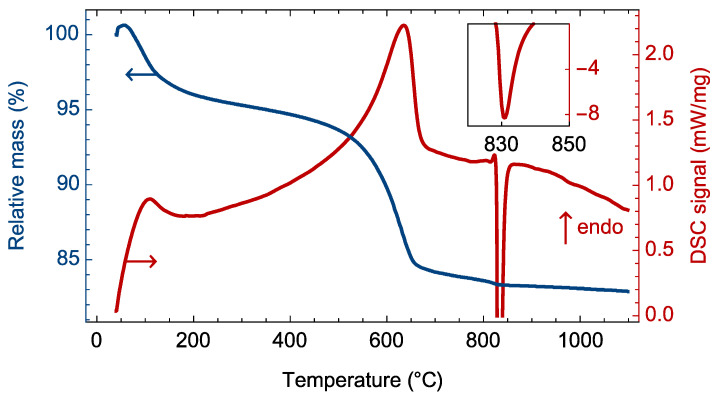
DSC-TG curves of the hydrothermal treatment product. The inset shows the peak of the sharp exothermic effect of magnesium silicate crystallization.

**Figure 3 materials-15-09023-f003:**
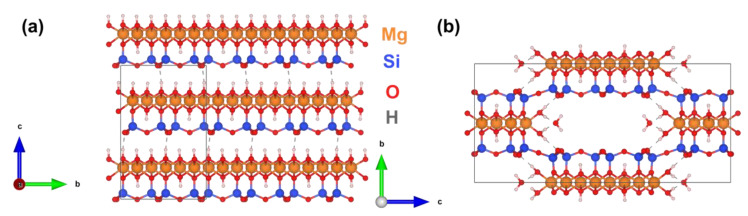
Crystal structures of (**a**) chrysotile and (**b**) sepiolite used in DFT calculations. Visualization was carried out using the VESTA 3 program [52]. The ‘b’ and ‘c’ letters denote crystallographic axes.

**Figure 4 materials-15-09023-f004:**
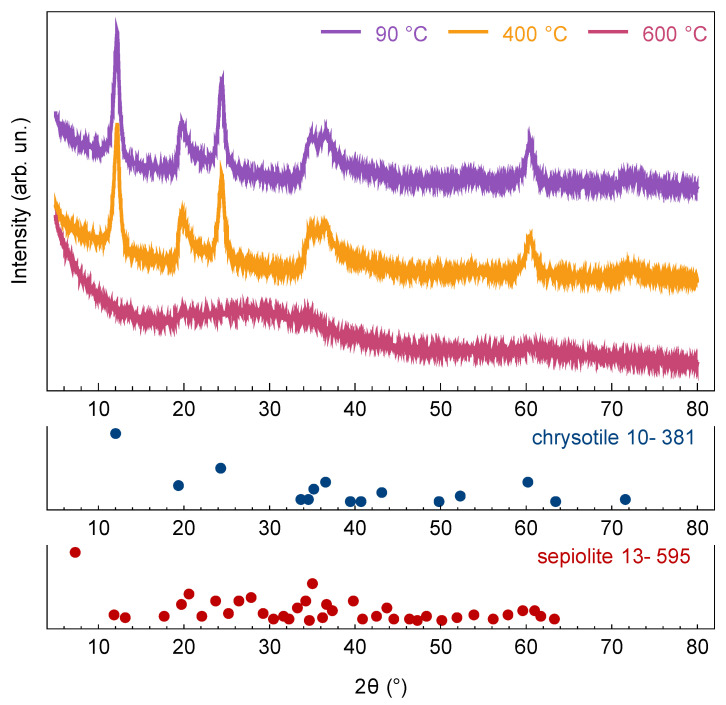
X-ray diffraction patterns and phase analysis of hydrothermal treatment products after drying at 90 °C and after heat treatment at 400 and 600 °C. Phase analysis was carried out using the ICDD PDF-2 database.

**Figure 5 materials-15-09023-f005:**
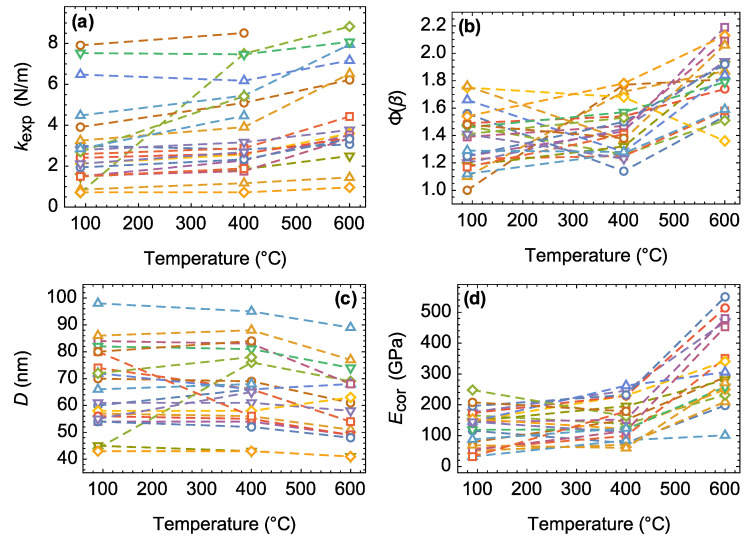
Changes in the main morphological and mechanical characteristics of individual nanobridges with an increase in heat treatment temperature: (**a**) bending stiffness, (**b**) Φ(β) parameter, (**c**) outer diameter, and (**d**) the Young’s modulus $E_{\mathrm{cor}}$. Each set of 3 points denotes the property alteration of the same nanobridge (see Figure 1).

**Figure 6 materials-15-09023-f006:**
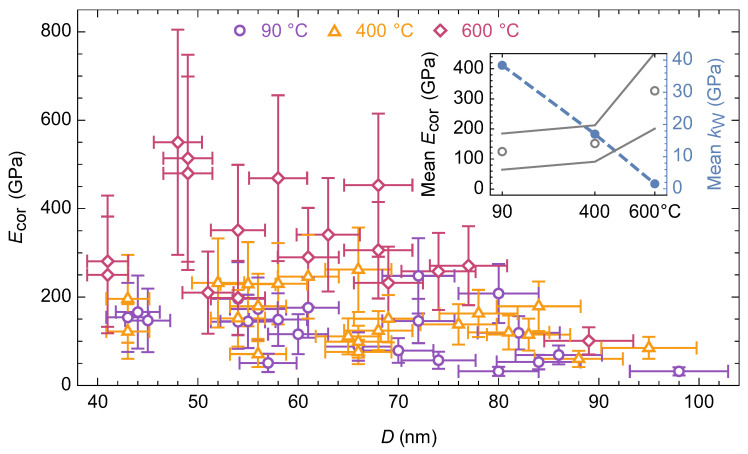
The Young’s modulus $E_{\mathrm{cor}}$ versus nanobridge outer diameter D after drying at 90 °C and after heat treatment at 400 and 600 °C. The inset shows the weighted average value of Young’s modulus versus the treatment temperature (colored areas mark one standard deviation), as well as the average value of the elastic foundation stiffness $k_{W}$.

**Figure 7 materials-15-09023-f007:**
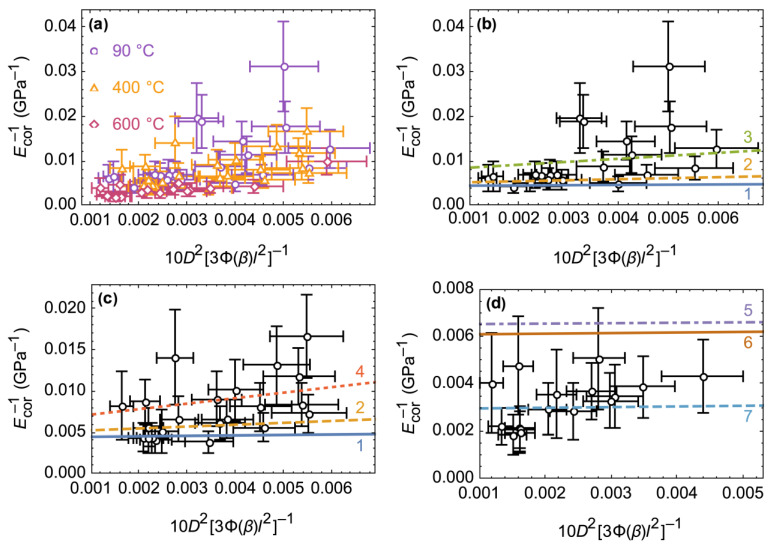
Linearization of the experimental data according to Equation (9) presented (**a**) in common range, and separately for (**b**) 90 °C, (**c**) 400 °C, and (**d**) 600 °C. The lines show the dependences calculated with Equation (9) using the $E_{G}$ and G values, respectively: 1–222 and 19 GPa; 2–196 and 4 GPa; 3–126 and 1.5 GPa; 4–152 and 1.5 GPa; 5–153 and 54 GPa; 6–164 and 40 GPa; 7–338 and 36 GPa. The values were taken from Table 1, measurement results (inset in Figure 6), as well as from previously published data [21,25], see text.

**Table 1 materials-15-09023-t001:** Lattice constants, Young’s modulus, and the shear modulus along different crystallographic directions of hydrosilicates with chrysotile and sepiolite structure, determined using DFT modeling.

Structure	Lattice Constants				$E_{\mathrm{DFT}}\;(\mathrm{GPa})$			$G_{\mathrm{DFT}}\;(\mathrm{GPa})$		
	a (Å)	b (Å)	c (Å)	*β* (*°*)	[100]	[010]	[001]	[100]	[010]	[001]
chrysotile	5.33	9.23	14.47	91.1	222	196	–	19	4	–
sepiolite	12.75	27.43	5.32	90	48	153	164	13	54	40

## Data Availability

Not applicable.

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
