# Peer review of "Thermal Treatment Impact on the Mechanical Properties of Mg3Si2O5(OH)4 Nanoscrolls"

_materials, 2022, doi:10.3390/ma15249023_

Round 1

Reviewer 1 Report

The manuscript focus on the study of mechanical properties evolution of synthetic phyllosilicate nanoscrolls after series of heat treatment. It is observed by an atomic force microscopy and calculated by density functional theory. The current manuscript needs some revisions. Here are some suggestions for modification:

1.     What are the advantages of choosing Mg3Si2O5(OH)4 nanoscrolls instead of other silicates as the research object in this paper, this should be elaborated in the Introduction.

2.     When performing density functional theory modeling, the author should give the specific size of the K-point sampling grid.

3.     On lines 188 and 189 of the manuscript, the author writes in the article "The k-point mesh was generated by the method of Monkhorst and Pack with a cutoff of 15 Å used for k-point sampling". This statement is somewhat vague, and 15 Å here should be the distance between the vacuum layers, which should be clearly described.

4.     On p.5, lines 195-197, the specific values of the lattice parameters of the unit cells of the model established by the author in this study should be given and compared with the previous calculated or experimental values to ensure the reliability of the established model.

5.     The significant digits after the decimal point of the data in Table 1 should be unified.

6.     The conclusion presented by the authors is rather a summary but does not give substantial insight into the role of thermal treatment impact on the mechanical properties of silicate materials.

Author Response

We would like to thank the review for the high evaluation of the manuscript.

Reviewer #1 wrote: What are the advantages of choosing Mg3Si2O5(OH)4 nanoscrolls instead of other silicates as the research object in this paper, this should be elaborated in the Introduction.

Our response: Among hydrosilicates, there are three main structures capable for spontaneous scrolling: Al2SiO3(OH)4imogolite, Al2Si2O5(OH)4 halloysite, and Mg3Si2O5(OH)4 chrysotile. First mineral forms mostly singlewalled nanotubes (about 2 nm in diameter) which are too hard to study. Halloysite is interesting, however, its synthesis is still complicated (we know only two papers, and nanoscrolls yield was questionable). Main complication originates from competition between strain and surface energy components for scrolling direction of halloysite layer (additional information could be found in paper https://doi.org/10.1007/s42860-020-00086-6). In contrast, chrysotile layer is free from that issue. Finally, we chose Mg3Si2O5(OH)4 composition because it is a sort of a standard to begin with if we consider some synthetic analogs like (Mg,Ni,Co,Fe)3(Si,Ti,Ge)2O5(OH)4.

Reviewer #1 wrote: 2. When performing density functional theory modeling, the author should give the specific size of the K-point sampling grid.

  1. On lines 188 and 189 of the manuscript, the author writes in the article "The k-point mesh was generated by the method of Monkhorst and Pack with a cutoff of 15 Å used for k-point sampling". This statement is somewhat vague, and 15 Å here should be the distance between the vacuum layers, which should be clearly described.

Our response: These two comments are related. There is no vacuum in our DFT models, the 3D crystals of chrysotile and sepiolite were studied. The k-grid cutoff in SIESTA is a parameter which determines the fineness of the k-grid used for Brillouin zone sampling. It is the half of the length of the smallest lattice vector of such a large supercell that is required to obtain the same sampling precision with a single k-point. In lines 194–196 of the revised manuscript we have added the original reference on technique of such k-grid generation in SIESTA: "...used for k-point sampling [Moreno, J,; Soler, J.M. Optimal meshes for integrals in real- and reciprocal-space unit cells. Phys Rev B 1992, 45, 13891-13898, doi: 10.1103/PhysRevB.45.13891]. Effectively, it corresponded to the k-point sampling grids 6´4´3 for chrysotile lattice and 3´2´6 for sepiolite lattice."

Reviewer #1 wrote:4. On p.5, lines 195-197, the specific values of the lattice parameters of the unit cells of the model established by the author in this study should be given and compared with the previous calculated or experimental values to ensure the reliability of the established model.

Our response: We have added several phrases devoted to lattice constants in rows 207–215 of the revised manuscript: "The equilibrium lattice parameters of both chrysotile and sepiolite demonstrate a fair reliability of the chosen DFT scheme for reproduction of the experimental lattice parameters within ±2% error. For chrysotile, theoretical data a = 5.33 Å, b = 9.23 Å, c = 14.47 Å, b = 91.1° fitted to experimental ones a = 5.340 Å, b = 9.241 Å, c = 14.689 Å, b = 93.66° [42]. For sepiolite, theoretical data a = 12.75 Å, b = 27.43 Å, c = 5.32 Å were comparable to experimental ones a = 13.405 Å, b = 27.016 Å, c = 5.275 Å [43], but the a-parameter was underestimated. Seemingly, this underestimation unveiled a swelling of real sepiolite by loose water molecules in channels of the lattice, which were skipped in our theoretical model."

Reviewer #1 wrote:The significant digits after the decimal point of the data in Table 1 should be unified.”

Our response: We have rounded off the Young’s and shear moduli values in Table 1.

Reviewer #1 wrote:The conclusion presented by the authors is rather a summary but does not give substantial insight into the role of thermal treatment impact on the mechanical properties of silicate materials.”

Our response: We have carefully evaluated the reviewer’s request and made some minor changes in the conclusion section. In our point of view, conclusion contains essential information about the mechanism of elastic properties alteration: a) before the heat treatment shear strain contributed a lot to the mechanical behavior of layered nanoscroll because adjacent layers were weakly bonded together; b) heat treatment initiated destruction of hydrogen bonds and formation of sepiolite-like phase with relatively strong Si–O–Si as demonstrated by thermal analysis and X-ray diffraction; c) as a result, shear strain became complicated.

Reviewer 2 Report

In this work, the authors studied the mechanical properties of Mg3Si2O5(OH)4 chrysotile nanoscrolls with atomic force microscopy and DFT calculations. The article has been well performed in a thorough scientific data. I have some minor comments:  

-          The application of the Mg3Si2O5(OH)4 should be added in the introduction.

-          The authors should give the reason for the different basis set for Mg, Si, H and O atoms in the DFT calculation.  The effect of temperatures should be added to compare with the experiment.

-          The elements in Figure 5 should be described. 

Author Response

We would like to thank the review for the high evaluation of the manuscript.

Reviewer #2 wrote: "The application of the Mg3Si2O5(OH)4 should be added in the introduction."

Our response: We have added some additional references, devoted to phyllosilicate application as catalyst supports and nanocontainers.

Reviewer #2 wrote: "The authors should give the reason for the different basis set for Mg, Si, H and O atoms in the DFT calculation. The effect of temperatures should be added to compare with the experiment."

Our response: The calculations of mechanical properties for a such large unit cell as the one for sepiolite (152 atoms) are time demanding. Therefore, a preliminary study of crystallographic and mechanical properties of chrysotile (72 atoms in unit cell) was performed using three basis sets - (a) the single-ζ basis for all elements, (b) the double-ζ basis for all elements and (c) the double-ζ basis for anionic O and the single-ζ basis for cationic Mg, Si, H. While all three sets were found capable for a satisfactory description of structural parameters and Young moduli, the set (a) demonstrated an unrealistic overestimation (>10 times) of shear moduli. The sets (b) and (c) yielded similar as well as reasonable values of shear moduli with a visible benefit in computational time for the set (c). Therefore, we add to the main text in lines 191–193 of the revised manuscript: "The chosen basis set is not inferior to the global double-ζ basis in description of crystallographic and mechanical properties of chrysotile with a visible benefit in computational time." Regrettably, none of the modern DFT methods is capable to trace the temperature dependence of mechanical and structural properties during degradation of chrysotile to sepiolite. The calculations can gather the data only for the initial (chrysotile) and the final states (a dry sepiolite) of such complex process.

Reviewer #2 wrote: " The elements in Figure 5 should be described."

Our response: We have added additional description to the Figure 5. Each set of three points was obtained by studying the same particles after their heat treatment, so each line should be described as ‘particle #1, particle #2 etc.’. We suppose that adding this legend would overload the figure.

Round 2

Reviewer 1 Report

I think the paper has been well revised.